

# Trends in N₂O and SF₆ mole fraction in archived air samples from Cape Meares, Oregon (USA) 1978–1996

Terry C. Rolfe[1] and Andrew L. Rice[1]

[1]Department of Physics, Portland State University, Portland, 97201, United States

*Correspondence to*: Terry C. Rolfe (trolfe@pdx.edu)

**Abstract.** Quantifying historical trends in atmospheric greenhouse gases (GHG) is important to understanding changes in their budgets and for climate modeling which simulates historic and projects future climate. Archived samples analyzed using updated measurement techniques and calibration scales can reduce uncertainties in historic records of GHG mole fractions and their trends in time. Here, we present historical measurements of two important GHG, nitrous oxide

($N_2O$) and sulfur hexafluoride ($SF_6$), collected at the midlatitude northern hemisphere station Cape Meares, Oregon (USA, 45.5° N, 124° W) between 1978 and 1996 in archived air samples from the Oregon Health and Science University – Portland State University (OHSU–PSU) Air Archive. $N_2O$ is the third most important anthropogenically forced GHG behind carbon dioxide ($CO_2$) and methane ($CH_4$). $SF_6$ has a low abundance in the atmosphere, but is one of the most powerful GHG known. Measurements of atmospheric $N_2O$ made during this period are available for select locations but prior to mid-1990 have

larger uncertainties than more recent periods due to advancements made in gas chromatography (GC) methods. Few atmospheric $SF_6$ measurements pre-1990 exist, particularly in the northern hemisphere. The GC system used to measure $N_2O$ and $SF_6$ concentrations in this work is designed to be fully automated, capable of running up to 15 samples per batch. Measurement precision ($1\sigma$) of $N_2O$ and $SF_6$ is 0.16% and 1.1% respectively. Samples were corrected for detector response non-linearity when measured against our reference standard, determined to be 0.14 ppb ppb$^{-1}$ in $N_2O$ and 0.03 ppt ppt$^{-1}$ in

$SF_6$. The concentration of $N_2O$ in archived samples is found to be 301.5 ± 0.3 ppb in 1980 and rises to 313.5 ± 0.3 ppb in 1996. The average growth rate over this period is 0.78 ± 0.03 ppb yr$^{-1}$ (95% CI). Seasonal amplitude is statistically robust, with a maximum anomaly of 0.3 ppb near April and a minimum near November of -0.4 ppb. Measurements of $N_2O$ match well with previously reported values for Cape Meares and other comparable locations. The concentration of $SF_6$ in analyzed samples is found to be 0.85 ± 0.03 ppt in 1980 and rises to 3.83 ± 0.03 ppt in 1996. The average growth rate over this period

is 0.17 ± 0.01 ppt yr$^{-1}$ (95% CI). Seasonality is statistically robust and has an annual peak anomaly of 0.04 ppb near January and a minimum anomaly of -0.03 ppt near July. These are unique $SF_6$ results from this site and represent a significant increase in $SF_6$ data available during the 1980s and early 1990s at any location. The concentration and growth rate of $SF_6$ measured compares well to other northern hemisphere measurements over this period. From these $N_2O$ and $SF_6$ measurements, overall we conclude that sample integrity is robust in the OHSU-PSU Air Archive.



# 1 Introduction

Anthropogenic sources of greenhouse gases (GHG) have altered the atmospheric composition resulting in a significant climate forcing near 3 W m$^{-2}$ since 1750 (Myhre et al. 2013). Measurements of GHG concentrations since the industrial revolution constrain global budget uncertainties and are central to interpret recent changes to source and sink

processes (Prinn et al. 2000; Khalil et al. 2002; Saikawa et al. 2014). When projecting future GHG concentrations, many additional factors must be included in models such as climate feedback effects and possible changes in transport processes. Uncertainties in model predictions can be minimized if GHG measurements are precise and span many different latitudes (Meinshausen et al. 2017).

When historical timeseries records are not available, past atmospheric GHG abundance can be evaluated using

archived air samples and ice core and firn air. One significant advantage of ice core and firn air for measuring past atmospheric concentrations of GHG is that samples may be collected today that represent past conditions. However, ice core and firn samples are difficult to obtain due to the remoteness of the locations where the samples are collected (Greenland and Antarctica) and provide limited spatial information. Temporal uncertainties also must be evaluated when measuring ice core and firn samples due to diffusion and gravitational separation (Ishijima et al. 2007); samples are best represented by a mean

age. By contrast, archived air samples are discrete in time and space, making them very valuable for evaluating past atmospheric abundance at specific periods in time. However, few air archives are available today. Archive samples may also contain storage artifacts that can contaminate historical records.

Nitrous oxide ($N_2O$) is the third most important GHG with anthropogenic sources after carbon dioxide ($CO_2$) and methane ($CH_4$). Today, the concentration of $N_2O$ is close to 330 ppb with a trend averaging 0.75 ppb yr$^{-1}$ over the last 30

years (Ciais et al., 2013). $N_2O$ has a large global warming potential (GWP), 298 times that of $CO_2$ over a 100-year period and a global radiative forcing estimated at 0.19 W m$^{-2}$ since 1750 (Myhre et al., 2013). The long lifetime (~120 years) results in most emitted $N_2O$ reaching the stratosphere, where photooxidation is the major source of stratospheric $NO_X$ ("active nitrogen"). $NO_X$ is the main natural catalyst of ozone ($O_3$) destruction (Crutzen 1970).

Anthropogenic sources of $N_2O$ account for roughly 40% of all $N_2O$ emissions, with natural sources accounting for

the other 60% (Ciais et al., 2013). Bottom-up calculations estimate anthropogenic production of 6.9 (2.7-11.1) TgN yr$^{-1}$ and natural production of 11 (5.4-19.6) TgN yr$^{-1}$. The uncertainty in these estimations is large, with 1σ error nearly ± 50%. Together with atmospheric measurements, top-down modelling better constrains the $N_2O$ budget and reduces uncertainty in the global source. Sources of $N_2O$ calculated this way estimate anthropogenic and natural source production of 6.5 (5.2-7.8) TgN yr$^{-1}$ and 9.1 (8.1-10.1) TgN yr$^{-1}$, respectively (Prather et al., 2012).

There are three major natural sources and six major anthropogenic sources of $N_2O$. Natural sources of $N_2O$ are natural soils (3.3-9.0 TgN yr$^{-1}$), oceans (1.8-9.4 TgN yr$^{-1}$), and atmospheric chemistry (0.3-1.2 TgN yr$^{-1}$) (Note: sources include the minimum and maximum estimates provided from bottom-up calculations in Ciais et al., 2013). By far, the largest anthropogenic source is agriculture, producing 1.7-4.8 TgN yr$^{-1}$, followed by industrial and fossil fuel sources (0.2-



1.8 TgN yr$^{-1}$), biomass burning (0.2-1 TgN yr$^{-1}$), rivers and estuaries (0.1-2.9 TgN yr$^{-1}$), atmospheric deposition (0.4-1.3 TgN yr$^{-1}$), and human excreta (0.1-0.3 TgN yr$^{-1}$) (Ciais et al. 2013). More constraints on source production provided through atmospheric measurements are needed to improve estimates of individual source magnitudes.

5     The main loss mechanism for $N_2O$ is destruction in the stratosphere through photolysis and the reaction with $O(^1D)$ (Prather et al. 2015). Soils and the oceans can act as sinks for $N_2O$ through microbial processes, however because the production of $N_2O$ is greater than what is consumed, the global net flux is positive. Estimates of the stratospheric sink account for 11.9 (11.0-12.8) TgN yr$^{-1}$ (Ciais et al., 2013).

    Rising global concentrations of $N_2O$ are due to the imbalance between the sources and the sinks. Based on a top-down constraint, the imbalance between sources and sinks is 3.6 (3.5-3.8) TgN yr$^{-1}$ (Ciais et al., 2013).

10     Models have shown that future climate conditions will likely amplify $N_2O$ production, meaning a linear increase in time may under-predict future concentrations based on the current rate of change (Khalil and Rasmussen 1983; Stocker et al. 2013). To minimize uncertainty in the $N_2O$ budget and in model projections, precise measurements of current and past atmospheric conditions from multiple global locations are needed. Measurements of atmospheric $N_2O$ made prior to mid-1990 have larger uncertainties than more recent periods due to advancements made in gas chromatography (GC) methods 15 (Prinn et al. 2000; Jiang et al. 2007). To reduce uncertainty during this period, archived samples may be analyzed using updated measurement techniques. Additionally, measurements of the isotopic composition of $N_2O$ in archived samples can constrain the $N_2O$ budget and changes in time due to characteristic isotopic effects in sources and sinks (Park et al. 2012; Snider et al. 2015).

    Sulfur hexafluoride ($SF_6$) is an extremely potent GHG with a GWP of 22800 (over 100 years compared to $CO_2$) and 20 a lifetime of ~3200 years (Ravishankara et al. 1993). While $SF_6$ is one of the strongest GHG controlled under emission regulations, it has a low global concentration (7.29 ppt in 2011), so it does not add significantly to climate forcing by itself (Myhre et. al. 2013).

    Sources of $SF_6$ are anthropogenic, with main uses being high voltage insulation, magnesium production and semiconductor manufacture (Maiss and Brenninkmeijer 1998; Olivier et al. 2005). Global production in 2008 was estimated 25 to be 7.16 Gg yr$^{-1}$ (Levin et al. 2010). With a very low solubility and no reactivity in the lower atmosphere, the only known sink for $SF_6$ is loss in the mesosphere.

    With almost all of the $SF_6$ that has been emitted since the industrial revolution to the atmosphere still present, global emissions can be accurately determined from observations of atmospheric concentration. Due to its long lifetime and anthropogenic origins, $SF_6$ is used as a validity check for atmospheric transport models (Levin and Hesshaimer, 1996; Patra 30 et al., 2009). It has been estimated that 94% of all $SF_6$ emissions originate in the northern hemisphere (Maiss et al. 1996), explaining a north-south hemisphere gradient in $SF_6$ concentration of about 0.4 ppt (Levin et al. 2010).

    Observations of the $SF_6$ growth rate have been reported by several studies (Levin et al. 2010; Rigby et al. 2010; Hall et al. 2011). The trend in $SF_6$ has varied over the last 30+ years and while the magnitude of the growth rate differs slightly between sample locations, several features are prominent. From the early 1970s to the mid-1990s, the trend steadily



increased from 0.1 ppt yr$^{-1}$ and peaked near 0.26 ppt yr$^{-1}$. The trend then slowly declined to ~0.20 ppt yr$^{-1}$ until the early 2000s, when the trend increased again. The inferred global emission of SF$_6$ from the trend increases nearly linearly from 2 Gg yr$^{-1}$ in the late 1970s to over 6 Gg yr$^{-1}$ in 1994-1995 (Levin et al. 2010; Rigby et al. 2010).

Reported atmospheric measurements of SF$_6$ before the year 1987 are few. In the southern hemisphere, Cape Grim, Tasmania (41° S, 145° E) archive measurements date back to 1978 (Levin et al. 2010). Northern hemisphere measurements are reported dating from 1973 from Trinidad Head, CA (41° N, 121° W), but few are prior to 1990 (Rigby et al. 2010). A more complete record of past SF$_6$ atmospheric concentrations is desirable.

The Oregon Health & Science University–Portland State University (OHSU-PSU) air archive includes archived air samples collected from Cape Meares, Oregon (45.5° N, 124.0° W) in the late 1970s, 1980s, and 1990s by the Department of Environmental and Bimolecular Systems, Oregon Graduate Institute of Science and Engineering (currently OHSU). The samples were collected by air liquefaction, where ~1000 L of air (STP) was compressed to 3000 kPa into 33 L electropolished stainless steel canisters. Today, archive samples are stored at Portland State University and contain pressures ranging from 60-2000 kPa (Rice et al. 2016). Here, we present details of the analytical technique employed and results from the analysis of 159 Cape Meares air samples from the OHSU-PSU air archive.

## 2 Methods

### 2.1 Gas chromatography analytical system

The gas chromatography (GC) analytical system (Fig. 1) employed at Portland State University for measuring N$_2$O and SF$_6$ in archived air samples is based on the configuration used by Hall et al. (2007) and references therein. We use an Agilent model 6890N gas chromatograph fitted with a micro-electron capture detector (μECD, Agilent Technologies, Santa Clara, CA). Peak separation is achieved by two Poropak Q 80/100 mesh columns (1.8 m × 2 mm i.d. pre-column, 3.7 m × 2 mm i.d. analytical column). The carrier gas is P5 (99.999%, Airgas, Portland, OR) equipped with O$_2$ and hydrocarbon traps (Restek, Bellefonte, PA) to further reduce impurities and found to significantly improve baseline signal stability. Two six-port switching valves (V$_1$ and V$_2$), a four-port switching valve (V$_3$), and a 16-port multi-position valve (Valvo Instrument Company Inc., Houston, TX) are controlled through Chemstation (V1.A, Agilent Technologies Inc., Santa Clara, CA).

A sample run begins in "back-flush" mode, with carrier gas flushing the pre-column in the reverse analytical direction to remove the build-up of water on the analytical column that would otherwise eventually elute to the μECD and affect signal baseline. A 16-port multi-position valve is used to introduce pressurized samples into the system; a 2-way electric valve (Clippard, Cincinnati, OH) is used to stop sample flow to the sample loop and prevent sample loss. Samples initially pass through a desiccant trap (Perma Pure, Toms River, NJ), before flushing a 10 ml sample loop at 60 ml min$^{-1}$ for 1.5 minutes. At this time, V$_3$ rotates, which places the system in "front-cut" mode and allows the sample loop to equilibrate. V$_1$ rotates at 1.75 minutes and allows the carrier gas to carry the sample N$_2$O and SF$_6$ to the pre-column where separation from O$_2$ and H$_2$O occurs. After O$_2$ elutes through the pre-column to vent, at 3 minutes V$_2$ rotates and places the pre-column



in line with the analytical column, transferring $N_2O$ and $SF_6$ to the analytical column. At 4.25 minutes, the sample has reached the analytical column and $V_1$, $V_2$, and $V_3$ rotate. This begins the back-flush of the pre-column while the analytes are carried to the µECD on the analytical column.

Oven and detector temperatures are maintained at 56° C and 310° C respectively. Carrier gas flow rates are 40 ml min$^{-1}$ maintained by electronic pressure control of the 6890N. $N_2O$ peak retention time is 6.1 minutes and $SF_6$ peak retention time is 7.0 minutes (Fig. 2). Peak integration is accomplished via Chemstation based on peak height.

All measurements of $N_2O$ and $SF_6$ are made relative to a calibrated whole air sample on the NOAA-06 $N_2O$ scale and NOAA-14 $SF_6$ scale (NOAA Tank CB11406-A, 328.71 ± 0.5 ppb $N_2O$, 8.76 ± 0.06 ppt $SF_6$), here-on referred to as the NOAA reference gas. Each sample is analysed 6 times and bracketed by 6 reference gas runs used to measure instrument
response and track signal drift. The GC-µECD analytical system was evaluated for precision, reproducibility, and linearity prior to its application to measure mole fraction in historic archive samples.

## 2.2 Precision and reproducibility of analytical system

Precision of measurement was determined by comparing residuals from sets of 6 gas analyses. Histogram distributions in figure 3 show 180 residuals (expressed as a percent relative standard deviation) collected from 30 sets of 6
measurements of $N_2O$ (Fig. 3a) and $SF_6$ (Fig. 3b) of the NOAA reference gas. Both $N_2O$ and $SF_6$ compare well to a normal distribution (black dashed lines), with chi-square goodness of fit p-values of 0.16 and 0.35, respectively. For $N_2O$, $1\sigma$ = 0.16% while for $SF_6$, $1\sigma$ = 1.1%. This corresponds to an uncertainty of ± 0.52 ppb for $N_2O$ and ± 0.10 ppt for $SF_6$.

Measurement reproducibility was evaluated by repeatedly measuring a dry air sample (Breathing Air, Airgas, Portland, OR) against the NOAA reference gas and evaluating consistency from the standard deviation of the results. The
sample was measured 18 times over a two-week period with mean measured concentrations of $N_2O$ and $SF_6$ of 390.9 ppb and 13.2 ppt, respectively. The standard deviations in $N_2O$ and $SF_6$ measurements are 0.46 ppb and 0.11 ppt respectively, which are indistinguishable from $1\sigma$ precision for a set of 6 NOAA reference gas measurements.

## 2.3 Linearity of the GC- µECD system

To ensure accurate results for this work, the detector response was evaluated over the mole fraction range expected
for $N_2O$ and $SF_6$ in the OHSU-PSU Air Archive. The range in northern hemisphere $N_2O$ mole fraction between 1978 and 1996 is between 295 and 314 ppb (Prinn et al. 2000; Ciais et al. 2013). Archived air sample measurements of northern hemisphere $SF_6$ mole fraction from Trinidad Head, CA measure below 1 ppt in the 1970s and rise to nearly 4 ppt in 1997; southern hemisphere measurements from Cape Grim, Tasmania and the South Pole show a similar range (Levin et al. 2010; Rigby et al. 2010).

A series of manometric dilutions were prepared from the NOAA reference gas at Portland State University to evaluate the µECD response over historical $N_2O$ and $SF_6$ mole fraction sample range. To characterize the $N_2O$ response, the





N$_2$O reference gas was diluted with ultra-pure air (zero grade, Airgas, Portland, OR; N$_2$O and SF$_6$ at concentrations below detection limits) using capacitance manometers (MKS Instruments, Andover, MA; range 0-10 torr and 0-1000 torr) into 3L electropolished stainless steel canisters (precision ± 0.01%). The range of N$_2$O concentrations produced in 3 L canisters was 32.2 - 321.4 ppb. Error introduced from the manometric process is small when compared to measurement uncertainty

(maximum 1σ error of ± 0.07 ppb for N$_2$O).

To characterize of the SF$_6$ response at low ppt concentrations requires consideration of the effect of the falling N$_2$O tail on the chromatogram baseline. To properly account for this interference, SF$_6$ dilutions at low concentrations (0.6 - 6.0 ppt) must have a N$_2$O concentration that reflects expected concentrations in archived samples (302–314 ppb). Prepared dilutions of SF$_6$ included the addition of an aliquot of 1 ppm N$_2$O (±5%, Scott Specialty Gases, St. Louis, MO) into the

canister prior to dilution with ultra-pure air. The maximum error (1σ) in SF$_6$ introduced from the manometric process is small (0.001 ppt) compared to measurement uncertainty. All dilution samples were measured at PSU on the GC-µECD system over several weeks to account for instrument drift. Tables 1 and 2 provides dilution sample pressures, calculated and observed µECD response, and measured N$_2$O and SF$_6$ mole fractions with the error in measurement used to characterize the GC-µECD linearity.

Results of linearity experiments are shown in figure 4. For N$_2$O, a slope of 0.870 ± 0.028 (95% CI) is found over the data range 289.7 - 328.7 ppb, most relevant for this work. A linear fit is a good model for the deviation from expected over this range (R$^2$ = 0.964); additional polynomial terms are not statistically robust. This results in sample measurements deviating from expected by ~0.14 ppb ppb$^{-1}$ N$_2$O difference from the NOAA reference. For the range of the N$_2$O in the OHSU-PSU air archive, all N$_2$O samples are adjusted for a linear correction of the form:

$$[N_2O]_X = a_1[N_2O]_Y + a_2 \qquad (1)$$

$$a_1 = 1.146 ± 0.037 \text{ (95\% CI)} \qquad (2)$$

$$a_2 = -47.95 ± 11.49 \text{ (95\% CI)} \qquad (3)$$

Where $[N_2O]_Y$ is the measured N$_2$O mole fraction and $[N_2O]_X$ is the corrected value. The slope and y-intercept, as well as their 95% confidence intervals, are represented by $a_1$ and $a_2$ respectively

The entire NOAA reference gas dilution range for N$_2$O (32 - 321 ppb) results in a deviation that can be adequately modeled using a 3$^{rd}$ degree polynomial. The linear fit discussed above is indistinguishable from the full 3$^{rd}$ degree polynomial over the N$_2$O concentration range of the OHSU-PSU Air Archive. However, if measuring N$_2$O samples with a difference of more than 80 ppb compared to the NOAA reference gas, the full 3$^{rd}$ degree polynomial is necessary to correct for the non-linear response in the µECD.

For SF$_6$, the prepared sample range over which the linear correction is applied is 0.59 – 8.76 ppt, most relevant for this work. The slope of the SF$_6$ linear fit is 0.971 ± 0.017 (95% CI) and is a good model for the deviation from expected over this range (R$^2$ = 0.9995). This results in a deviation from expected of ~0.03 ppt ppt$^{-1}$ SF$_6$ difference from the NOAA reference when measuring samples. All SF$_6$ measurements are adjusted for a linear correction of the form:

$$[SF_6]_X = b_1[SF_6]_Y + b_2 \qquad (4)$$




$b_1 = 1.03 \pm 0.018$ *(95% CI)* (5)

$b_2 = -0.297 \pm 0.099$ *(95% CI)* (6)

Where $[SF_6]_Y$ is the measured $SF_6$ mole fraction and $[SF_6]_X$ is the corrected value. The slope and y-intercept, as well as their 95% confidence intervals, are represented by $b_1$ and $b_2$, respectively.

Detector response non-linearity has been evaluated in previous work by other groups on GC-ECD systems. For $N_2O$, deviations from expected of $\sim 0.2$ ppb ppb$^{-1}$ difference from the reference gas are reported (Schmidt et al. 2001; Hall et al. 2007). These are similar to the value reported here for the μECD. Over larger ranges, a similar non-linear response curve is also reported. $SF_6$ non-linearity reported in Levin et al. (2010) has a similar curvature to the full $N_2O$ non-linear response found here and previously discussed. Yet, this curvature is not observed to be significant over the range of $SF_6$ dilutions

conducted here.

## 3 Results and Discussion

### 3.1 Air archive mole fractions of $N_2O$ and $SF_6$

Measurements of $N_2O$ and $SF_6$ mole fraction from 159 samples of the OHSU-PSU Air Archive were initially filtered for analysis using a 7 median absolute deviation (7MAD) noise filter to remove far outliers. Polynomial fits (1$^{st}$

degree for $N_2O$ and 2$^{nd}$ degree for $SF_6$) were then applied to the data. Residual values outside of $2\sigma$ for $N_2O$ and $3\sigma$ for $SF_6$ were removed for further data analysis. The entire process removed 12 data points for $N_2O$ and 4 data points for $SF_6$ used in analysis.

Deseasonalized measurements of $N_2O$ and $SF_6$ from Cape Meares are shown in figure 5a and 5b, respectively. A locally weighted linear regression (LOWESS) is used to smooth the data using a 3-year smoothing window (Cleveland and

Devlin 1988). The confidence intervals around regressions are calculated by bootstrapping residual variability 1000 times. The regression results in a concentration of $301.5 \pm 0.3$ ppb ($1\sigma$) in 1980 and increasing roughly linearly to the mid 1990s, where the concentration is $313.5 \pm 0.3$ ppb ($1\sigma$) in 1996.

Observations of $N_2O$ mole fraction match well with previously published measurements of $N_2O$ from Cape Meares between 1978 and 1998 of 301.2 ppb in 1980 and 313 - 314.5 ppb in 1996 on the SIO-1998 $N_2O$ scale (Prinn et al. 1990;

Prinn et al. 2000; Khalil et al. 2002). The $N_2O$ scale difference between SIO-1998 and NOAA-06 is minimal (Hall et al. 2007). Additional measurements by the Advanced Global Atmospheric Gases Experiment (AGAGE) and NOAA/ESRL (on the SIO-1998 $N_2O$ and NOAA-06 $N_2O$ scales, respectively) are reported from comparable sample locations. Trinidad Head, CA (41° N, 121° W), Mace Head, Ireland (53° N, 10° W), and Niwot Ridge, CO (40° N, 106° W) all measure ~313 ppb in 1996 (Prinn et al 2000; Hall et al. 2007). Together, these comparisons indicate the $N_2O$ in the archived samples has stored

well.

Measured $SF_6$ concentration in archived Cape Meares samples is determined to be $0.85 \pm 0.03$ ppt ($1\sigma$) in 1980 and increases to a concentration of $3.83 \pm 0.03$ ppt ($1\sigma$) in 1996. Cape Meares does not have previously reported measurements





of SF$_6$ to compare with directly. Measurements of SF$_6$ from Trinidad Head, CA are reported to be ~0.85 ppt in 1980 and ~3.73 ppt in 1996 on the SIO-2005 SF$_6$ scale (Rigby et al. 2010). To convert to the NOAA-06 SF$_6$ scale, values measured on the SIO-2005 SF$_6$ scale are divided by a conversion factor of 0.9991 (Hall et al. 2014). In 1996, values of 3.87 ppt, 3.87 ppt, and 3.78 ppt are reported for Alert, Canada (82° N, 62° W), Barrow, AK (71° N, 157° W), and Niwot Ridge, CO

respectively on the NOAA-06 SF$_6$ scale (Hall et al. 2011). At these SF$_6$ concentrations, the difference between the NOAA-06 scale and the NOAA-14 scale is minimal. Cape Meares SF$_6$ measured values compare well with these northern hemisphere locations.

In the northern hemisphere, maximum background concentration measurements of SF$_6$ are reported from mid-to-high latitudes. For the year 1994, measurements from Fraserdale, Canada (50° N, 82° W) are reported to be 0.14 ppt higher

than samples measured from Izaña, Tenerife (28° N, 16° W) (Maiss et al. 1996). This difference is explained by the vast majority of SF$_6$ emissions coming from the mid-latitudes in the northern hemisphere (Maiss and Brenninkmeijer 1998; Levin et al. 2010; Rigby et al. 2010). The measured SF$_6$ concentrations from Cape Meares, also a midlatitude NH site, appear to fit in well with the expected meridional gradient when comparing to previously mentioned reported values.

Southern hemisphere measurements of SF$_6$ from archived atmospheric samples from Cape Grim, Tasmania (41° S,

145° E) and Neumayer, Antarctica (70° S, 8° W) are ~0.6 – 0.7 ppt in 1980 and ~3.4 – 3.5 ppt in 1996 on SIO-2005 and University of Heidelberg SF$_6$ scales (Levin et al. 2010, Rigby et al. 2010). As with the SIO-2005 SF$_6$ scale, the NOAA-06 and University of Heidelberg scale differences are small. To convert to the NOAA-06 SF$_6$ scale, values measured on the University of Heidelberg SF$_6$ scale are divided by a conversion factor of 0.9954 (Hall et al. 2014). Including a scale correction, Cape Meares SF$_6$ measurements are higher than Cape Grim and Neumayer during this period by 0.2 - 0.4 ppt.

Much or all of this difference can be explained by an interhemispheric north-south difference of 0.3 – 0.4 ppt (Levin et al. 2010).

## 3.2 Growth rate in N$_2$O and SF$_6$

The mean secular trend between 1978 and 1996 for N$_2$O and SF$_6$ is 0.78 ± 0.03 ppb yr$^{-1}$ (95% CI) and 0.17 ± 0.01 ppt yr$^{-1}$ (95% CI) respectively, determined by applying a linear fit to deseasonalized data. These trends translate to annual

increases of ~ 0.25% yr$^{-1}$ and ~ 10% yr$^{-1}$ for N$_2$O and SF$_6$, respectively. Annual trends for N$_2$O and SF$_6$ at Cape Meares, Oregon are determined from the derivative of the deseasonalized localized regression (Fig. 5 c&d). Uncertainty bands are generated from regressions of bootstrapped variability. Data points represent the mean annual trend with error bars equal to ±1σ of the trend over the year.

The mean annual trend in N$_2$O (Fig. 5c) ranges between 0.6 ppb yr$^{-1}$ and 1.2 ppb yr$^{-1}$. All years between 1980 and

1996 show a positive rate of change significant at the 95% confidence level. The uncertainty in the annual trend is smallest in the early 1980s, at ± 0.15 ppb yr$^{-1}$ (95% CI), where there are largest numbers of data (~50% of samples are between 1980



and 1985). After 1985, uncertainty in the annual trend becomes $\pm$ 0.5 ppb yr$^{-1}$ (95% CI). This relatively large uncertainty results in an annual growth rate that is statistically indistinguishable between years.

A previously reported secular trend of $N_2O$ reported between 1978 and 1998 for Cape Meares is 0.74 $\pm$ 0.02 ppb yr$^{-1}$, indistinguishable from our result (Prinn et al. 2000). The global secular trend of $N_2O$ for the period 1985 to 1996 reported

by Khalil et al. (2002) is 0.69 $\pm$ 0.03 ppb yr$^{-1}$, also compatible with our trend at Cape Meares.

The $SF_6$ annual trend (Fig. 5d) from the Cape Meares analysis increases from 0.07 $\pm$ 0.03 ppt yr$^{-1}$ (95% CI) in 1980 to 0.26 $\pm$ 0.05 ppt yr$^{-1}$ (95% CI) in 1994. The average rate of change in the growth rate (second derivative of mole fraction vs. time) over this period is 0.014 ppt yr$^{-2}$. The increase in growth rate over this period is statistically significant at high levels of confidence (95%). After 1994, we measure a decrease in the growth rate, though this decline is not statistically

significant at high levels of confidence over this short time interval.

Comparable trends in $SF_6$ measured at other locations are available for the mid-1990s. The average global growth rate of $SF_6$ in 1994 was reported at 0.23 ppt yr$^{-1}$ in the northern hemisphere (Maiss et al. 1996). Alert, Canada and Izaña, Tenerife are observed to have maximum trends of 0.26 ppt yr$^{-1}$ in mid-1994 and at the beginning of 1995, respectively, compatible with results presented here (Levin et al. 2010). The localized maximum in growth rate in ~1994 observed here is

present in some southern hemisphere observations of $SF_6$ at a similar time as well; Neumayer, Antarctica shows a maximum trend in 1995-1996 of 0.25 ppt yr$^{-1}$ (Levin et al. 2010). This finding is consistent with a peak in $SF_6$ emissions as reported by the European Database for Global Atmospheric Research (EDGAR, v4.2).

Another feature observed in the $SF_6$ trend from Cape Meares is a local maximum in the growth rate near 1987 (Fig. 5d). Notably however, not all data sets agree. The growth rate reported from Neumayer, Antarctica has this feature during a

similar period (Levin et al. 2010), but the trend reported at Cape Grim, Tasmania does not show this local maximum (Rigby et al. 2010). Due to the large uncertainty from the few archived samples available during that time period, this local maximum is not statistically distinguishable from surrounding years at high levels of confidence in the Cape Meares analysis and this result is merely suggestive. Additional evidence is needed to corroborate this finding.

### 3.3 Seasonality in $N_2O$ and $SF_6$ mole fraction

Seasonal behaviour for $N_2O$ and $SF_6$ are shown in figure 6 determined from residuals to the secular trend. The $N_2O$ seasonal cycle at Cape Meares shows a maximum near April and May of 0.3 ppb and an extended minimum from September through December of –0.4 ppb. Although there is considerable uncertainty surrounding monthly means, the difference between the spring maximum and fall minimum is statistically robust at high levels of confidence (2-sample KS test p-value = 0.003).

The seasonal amplitude matches well with previously reported northern hemisphere magnitudes of $\pm$ 0.4 ppb (Liao et al. 2004). Seasonal phase is also similar to Cape Meares at other mid-latitude northern hemisphere sites. $N_2O$ seasonality reported at Mace Head, Ireland has a maximum near April and a minimum near August and September (Nevison et al. 2004;



Jiang et al. 2007) and Trinidad Head, CA seasonality has a maximum near late May and a broad minimum from September to January (Nevison et al. 2007).

In general, $N_2O$ seasonal amplitude is known to vary strongly with latitude, e.g., 0.29 ppb at the South Pole (90° S, 102° W) and 1.15 ppb at Alert, Canada (Jiang et al. 2007). This is attributed in part to the stronger branch of the Brewer
Dobson circulation in the northern hemisphere which also explains the high latitude minimums in late-summer months related to the influx of $N_2O$ depleted air from the stratosphere during the spring (Liao et al. 2004; Nevison et al. 2004). Aside from atmospheric circulation, $N_2O$ seasonality may also be influenced by regional sources. Leuker et al. (2003) suggested local maximums at Trinidad Head may reflect the influence of strong coastal upwelling. Similarly located in the Eastern Pacific, Cape Meares may also be subject to coastal upwelling influences. Isotopic analysis or modeling of transport effects
and source influence would be useful to help interpret seasonal behaviour of $N_2O$ at Cape Meares.

Seasonality for $SF_6$ shows a maximum between December and February of 0.04 ppt and a minimum near July of - 0.03 ppt. The difference between the winter maximum and summer minimum is statistically significant (2-sample KS test p-value = 0.004). $SF_6$ seasonality has not previously been reported for Cape Meares.

Some seasonality in northern hemisphere observations of $SF_6$ is reported in the literature at select locations. Barrow,
AK has a minimum in September and October with a broad maximum from December to June (Patra et al. 2009). Alert, Canada shows a strong minimum in October, though a maximum is not clearly defined (Wilson et al. 2014). Continental sites such as Niwot Ridge show large interannual variability (IAV) but have little distinguishable seasonality (Patra et al. 2009).

$SF_6$ seasonality at Cape Grim has been reported to have amplitude of ± 0.01 ppt with a maximum in September and
October and a minimum in near February (Nevison et al. 2007; Wilson et al. 2014). The seasonal phase of Cape Grim is nearly anti-phase of Cape Meares reported here, though the amplitude is a factor of 4 smaller at Cape Grim. Similar to $N_2O$, seasonal amplitude is expected to be larger in the northern hemisphere than in the southern hemisphere (Nevison et al. 2007). Because sources of $SF_6$ are a-seasonal and sinks are essentially zero in the troposphere, the driving force behind the observed seasonality in $SF_6$ is considered to be atmospheric transport. Processes such as convection, vertical diffusion, boundary layer
mixing, and shifts in the ITCZ can potentially influence the observed seasonality at a location. Modeling atmospheric transport effects on $SF_6$ at Cape Meares could help confirm amplitude and phase reported here.

## 4 Conclusions

We have measured 159 samples from the OHSU-PSU Air Archive collected at Cape Meares, Oregon (45.5° N, 124.0° W) for $N_2O$ and $SF_6$ mole fraction using GC-µECD spanning April 1978 to December 1996. The GC-µECD system is
designed to be fully automated, capable of running multiple pressurized samples per run. Measurement precision of $N_2O$ and $SF_6$ is 0.16% and 1.1% respectively. Sample concentrations were corrected for detector response non-linearity when





measured against our reference gas. The linearity correction was found to be 0.14 ppb ppb$^{-1}$ and 0.03 ppt ppt$^{-1}$ for N$_2$O and SF$_6$, respectively.

Analysis of archived air samples finds the mole fraction of N$_2$O in 1980 to be 301.5 ± 0.3 ppb (1σ) and rises to 313.5 ± 0.3 ppb (1σ) in 1996. The average growth rate over this period is 0.78 ± 0.03 ppb yr$^{-1}$ (95% CI). Seasonality shows

peak amplitude of 0.3 ppb near April and minimum amplitude of -0.4 ppb near November and is statistically robust. Our measurements of N$_2$O were found to match well with previously reported values for Cape Meares and other comparable northern hemisphere mid-latitude locations.

For SF$_6$, the concentration in 1980 is found to be 0.85 ± 0.03 ppt (1σ), increasing to 3.83 ± 0.03 ppt (1σ) in 1996. The average growth rate over this period is 0.17 ± 0.01 ppt yr$^{-1}$ (95% CI). Seasonality shows peak amplitude of 0.04 ppb

near January and minimum amplitude of -0.03 ppt near July. There are no previous reported measurements of SF$_6$ from Cape Meares to compare against directly. SF$_6$ measurements compare well to other northern hemisphere measurements from Levin et al. (2010), Rigby et al. (2010), and Hall et al. (2011) over similar time periods when including spatial variability. From these N$_2$O and SF$_6$ measurements, we can conclude the sample integrity is robust within the OHSU-PSU Air Archive from Cape Meares, Oregon. Resulting dataset of SF$_6$, in particular, contributes to a better characterization of historic SF$_6$

growth rate and its atmospheric variability over this period of dramatic growth.

## Competing interests

The authors declare that they have no conflict of interest.

## Author Contribution

Both authors (Terry Rolfe and Andrew Rice) worked closely together in the development and implementation of the GC

technique used to make the mole fraction measurements of N$_2$O and SF$_6$. Data curation and analysis was also completed by both authors. The original draft of this manuscript was prepared by Terry Rolfe with Andrew Rice responsible for the review and editing of the manuscript.

## Acknowledgements

This study was supported by the US National Science Foundation (Atmospheric and Geospace Sciences grant 0952307). The

authors would like to recognize Johnathan Radda for his assistance developing the GC-μECD technique used to measure the samples, R. A. Rasmussen for help establishing the OHSU-PSU air archive, and Christopher Butenhoff and M. A. K. Khalil for their assistance with data analysis and interpretation.





*Data availability:* A supplementary dataset of N$_2$O and SF$_6$ mole fractions at Cape Meares, Oregon measured for this work from the OSHU-PSU Air Archive are available to the scientific community (upon publication) and may be obtained by contacting the corresponding author.

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



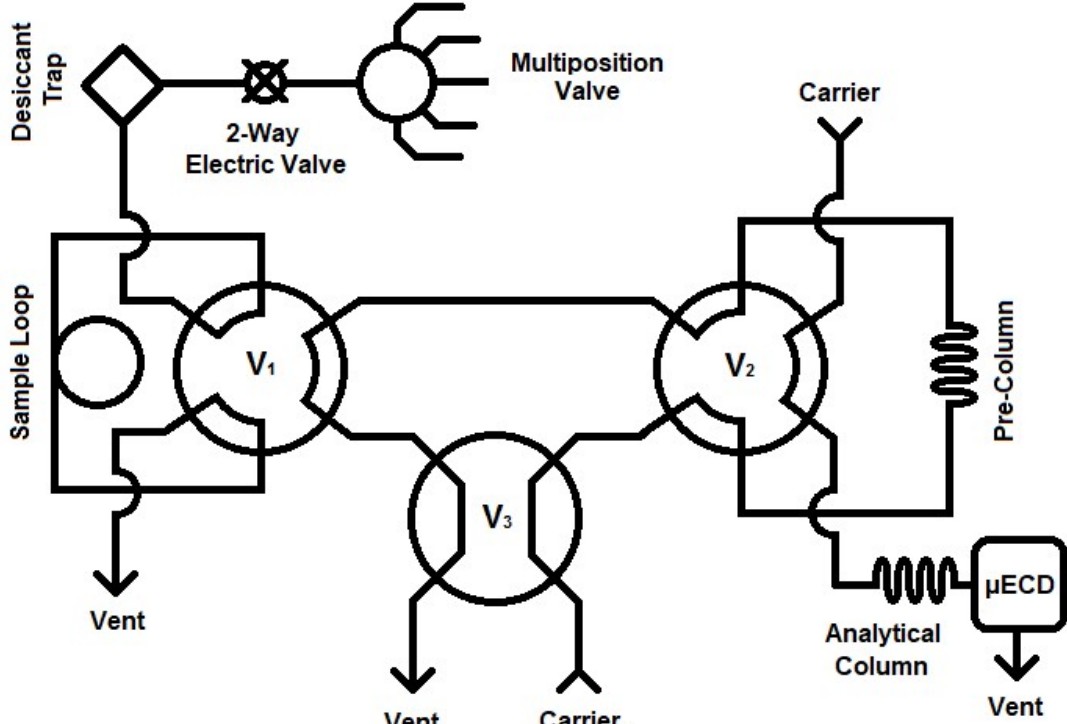

**Figure 1.** Schematic view of the analytical system for sample evaluation. The system is shown in "back-flush" mode. $V_1$ = Valve 1, $V_2$ = Valve 2, $V_3$ = Valve 3.





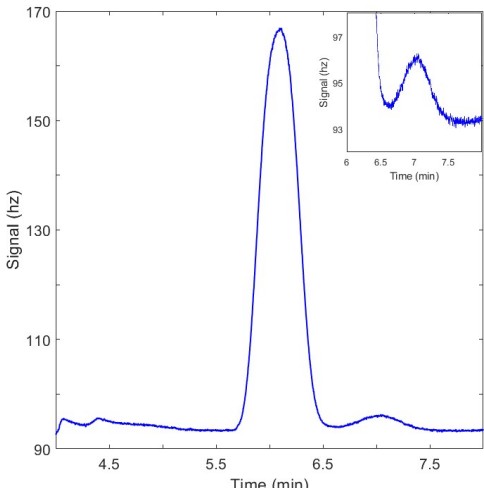

**Figure 2.** Sample chromatogram showing $N_2O$ peak at a retention time of 6.1 minutes and $SF_6$ peak at a retention time of 7.0 minutes. Upper-right corner inlay shows an enlarged plot of the $SF_6$ peak.

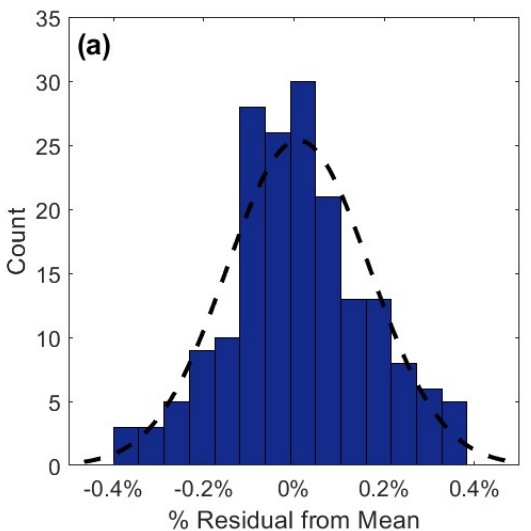
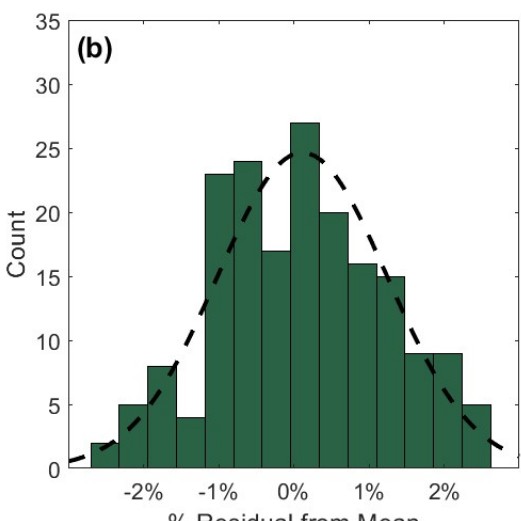

**Figure 3.** Precision in measurement for $N_2O$ (a) and $SF_6$ (b) expressed as percent relative standard deviation from 30 sets of 6 measurements of the NOAA reference gas. The black dotted line represents a normal distribution curve with the same mean and standard deviation. The standard deviation for $N_2O$ and $SF_6$ is 0.16% and 1.1%, respectively.



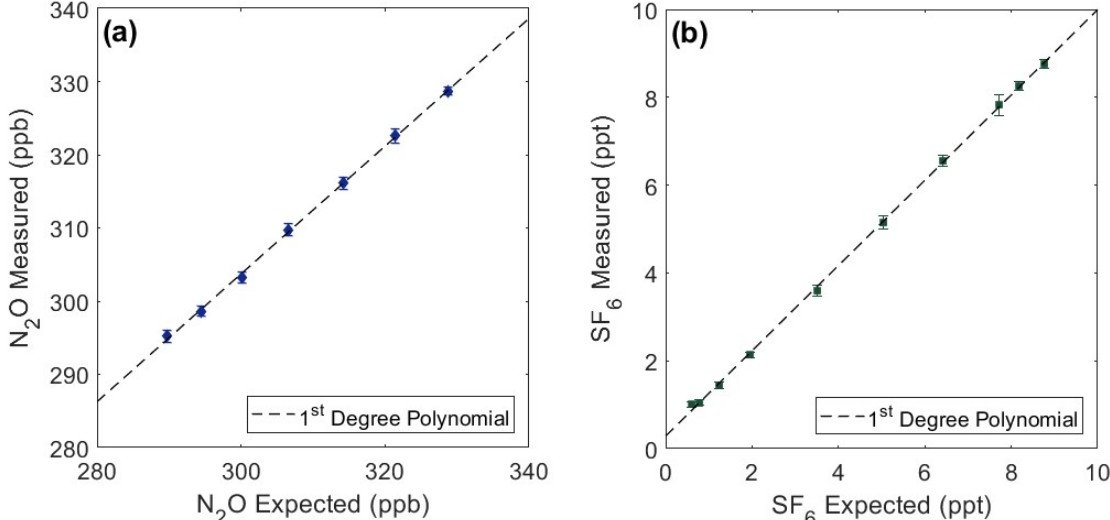

**Figure 4.** Measurement linearity from plots of measured mole fraction vs. expected mole fraction of $N_2O$ (a) and $SF_6$ (b). Expected mole fraction is calculated from the NOAA reference mole fraction (328.71 ppb $N_2O$ and 8.76 ppt $SF_6$) after dilution with ultra-pure air. Error bars represent $1\sigma$ total uncertainty.



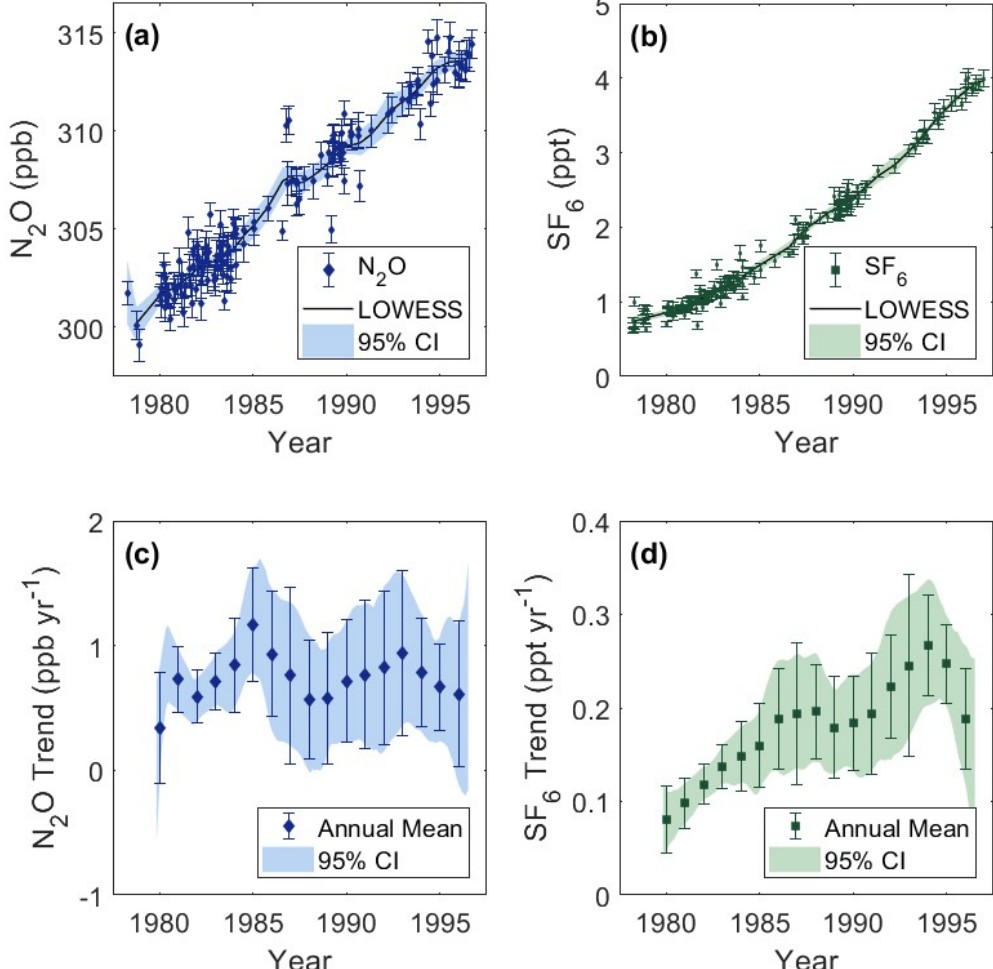

**Figure 5.** Deseasonalized measurements of mole fraction versus date of collection, $N_2O$ (a) and $SF_6$ (b), and annual trends in time from Cape Meares, Oregon, $N_2O$ (c) $SF_6$ (d). Error bars are $1\sigma$ uncertainty. The solid black lines are LOWESS fit to the data using a smoothing window of 3 years and shaded areas are 95% confidence intervals in the LOWESS fit calculated 5   from bootstrapping residual variability 1000 times.



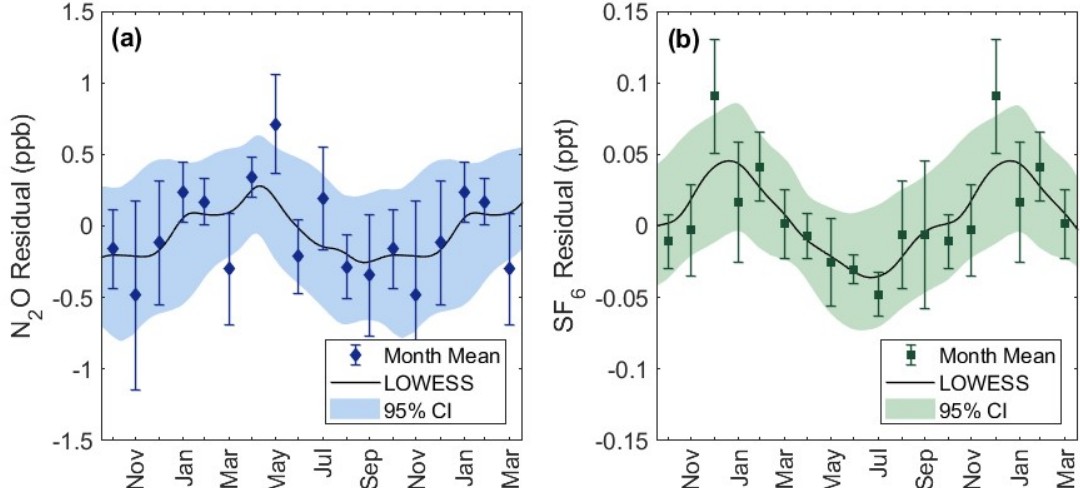

**Figure 6.** Seasonality for $N_2O$ (a) and $SF_6$ (b) calculated from the residuals of observed data points to the secular trend. The black line is a LOWESS fit to residuals with a smoothing window of 1 month. Data points show observed monthly mean residual after binning by month with error bars representing standard error within the month. Shaded areas are 95% CI
5    calculated from 1000 bootstrapped LOWESS fits while including the measurement uncertainty to each data point.



**Table 1.** Characteristics of 12 manometric $N_2O$ dilution samples prepared at Portland State University.

| Canister ID | a. $P_{Ref}$ (kPa) | b. $P_{Total}$ (kPa) | c. Expected $N_2O$ Response | d. Measured $N_2O$ Response | e. Measured $N_2O$ (ppb) | f. $N_2O$ $1\sigma$ (ppb) |
|---|---|---|---|---|---|---|
| 1.7 | 97.0 | 132.4 | 0.7327 | 0.7673 | 252.21 | 0.78 |
| 1.14 | 36.6 | 132.2 | 0.2767 | 0.3262 | 107.22 | 0.45 |
| 1.5 | 61.5 | 132.9 | 0.4627 | 0.5157 | 169.52 | 0.51 |
| 2.14 | 21.0 | 132.6 | 0.1585 | 0.1941 | 63.79 | 0.27 |
| 2.7 | 13.0 | 132.5 | 0.0978 | 0.1239 | 40.73 | 0.44 |
| 2.5 | 80.5 | 132.2 | 0.6092 | 0.6535 | 214.80 | 0.52 |
| 3.7 | 127.0 | 132.8 | 0.9559 | 0.9618 | 316.15 | 0.83 |
| 3.5 | 117.0 | 132.8 | 0.8813 | 0.8981 | 295.20 | 0.87 |
| 3.14 | 123.8 | 132.7 | 0.9326 | 0.9423 | 309.75 | 0.85 |
| 4.5 | 129.7 | 132.7 | 0.9778 | 0.9813 | 322.56 | 0.96 |
| 4.14 | 119.1 | 132.9 | 0.8959 | 0.9085 | 298.62 | 0.71 |
| 4.7 | 120.9 | 132.5 | 0.9129 | 0.9226 | 303.26 | 0.80 |

**a.** $P_{Ref}$ is the NOAA reference gas pressure (in kPa) introduced to the canister.

**b.** $P_{Total}$ is the final pressure (in kPa) of the canister after balancing with ultra-pure air.

**c.** Expected response is calculated from the $P_{Ref}/P_{Final}$ fraction.

**d.** Measured $N_2O$ response of the µECD.

**e.** Measured $N_2O$ in ppb.

**f.** $N_2O$ $1\sigma$ (ppb) is from combined uncertainty of sample and surrounding NOAA reference.

**Table 2.** Characteristics of 9 manometric $SF_6$ dilution samples prepared at Portland State University.

| Canister ID | a. $P_{Ref}$ (kPa) | b. $P_{Scotty}$ (kPa) | c. $P_{Total}$ (kPa) | d. Expected $SF_6$ Response | e. Measured $SF_6$ Response | f. Measured $SF_6$ (ppt) | g. $SF_6$ $1\sigma$ (ppt) |
|---|---|---|---|---|---|---|---|
| 1.14 | 97.0 | - | 132.4 | 0.7327 | 0.7476 | 6.55 | 0.12 |
| 3.5 | 117.0 | - | 132.8 | 0.8812 | 0.8943 | 7.83 | 0.24 |
| 3.14 | 123.8 | - | 132.7 | 0.9326 | 0.9414 | 8.25 | 0.10 |
| 1.1 | 29.6 | 31.0 | 132.6 | 0.2230 | 0.2443 | 2.14 | 0.06 |
| 1.18 | 11.9 | 36.9 | 133.0 | 0.0896 | 0.1199 | 1.05 | 0.07 |
| 1.28 | 8.9 | 37.6 | 132.3 | 0.0674 | 0.1153 | 1.01 | 0.06 |
| 2.1 | 18.7 | 34.8 | 131.9 | 0.1418 | 0.1644 | 1.44 | 0.08 |
| 2.18 | 75.9 | 16.0 | 132.2 | 0.5740 | 0.5879 | 5.15 | 0.15 |
| 2.28 | 52.9 | 23.5 | 132.1 | 0.4002 | 0.4110 | 3.60 | 0.12 |

**a.** $P_{Ref}$ is the NOAA reference gas pressure (in kPa) introduced to the canister.

**b.** $P_{Scott}$ is the 1 ppm $N_2O$ balanced with He (in kPa) introduced to the canister.

**c.** $P_{Total}$ is the final pressure (in kPa) of the canister after balancing with ultra-pure air.

**d.** Expected $SF_6$ response is calculated from the $P_{Ref}/P_{Final}$ fraction.

**e.** Measured $SF_6$ response of the µECD.



**f.** Measured SF$_6$ in ppt.

**g.** SF$_6$ 1σ (ppt) is from combined uncertainty of sample and surrounding NOAA reference.