# Peer review of "Trends in $N_2O$ and $SF_6$ mole fraction in archived air samples from Cape Meares, Oregon (USA) 1978–1996"

_Atmospheric Chemistry and Physics, 2019_

## Referee Comment (RC1) · Eric Ray (Referee) · 5 Mar 2019

This paper describes new measurements of SF6 and N2O from archived air samples taken over two decades ago from a site in Oregon. Measurements of these trace gases from before the mid 1990s are relatively rare so this addition is valuable, especially for SF6, of which there are very few measurements going back to the 1970s. The description of the measurement and calibration techniques are thorough and well done. The features of the measurement time series and seasonal cycles are interesting and described well. My main comment is that it would be helpful to see a plot of the comparison with other measurements of these trace gases rather than in the text only. The

subject is appropriate for ACP and I recommend publication with consideration of the minor comments below.

Specific comments:

Pg. 1, line 25: I think you mean 0.04 ppt rather than ppb.

Pg. 3, line 19-20: There are two more recent studies on the lifetime of SF6 that should be included here since they both significantly reduce the estimated lifetime, Ray et al., JGR, 2016 and Kovacs et al., ACP, 2017.

Pg. 6, line 12: Change 'provides' to 'provide'

Pg. 7, line 21: Even though it's apparent from the values of the concentration you should add 'N2O of' before '301.5' since the figure includes both N2O and SF6.

Section 3.1: You mention comparable measurements and their locations in the text of this section but it would be easier to see this information in a figure. What would be useful is a plot of concentration vs. latitude at two different times, one at the beginning of your measurement time series and one at the end. By including all available surface measurements it will be easy to see how many other measurements exist for each time and how it changed. Since the concentrations changed enough over the period of your measurements you could just color the two different times differently and they will fit on the same plot. The lack of measurements in the 1970s should be readily apparent from a plot of this type.

Pg. 10, lines 23-25: Also, seasonal transport from the stratosphere can influence SF6 due to the high growth rates, especially in these early years. Growth rates of $\sim$10%/yr means that stratospheric air with a mean age of 2 years will have $\sim$20% lower concentrations compared to tropospheric values. For example, the seasonal cycle of CFCs have a minimum in the summer of each hemisphere due to the transport of relatively low concentrations due to photochemical destruction (e.g. Liang et al., JGR, 2008).

---

## Referee Comment (RC2) · Brad Hall (Referee) · 15 Mar 2019

Trends in N2O and SF6 mole fraction in archived air samples from Cape Meares, Oregon (USA) 1978–1996; acp-2019-114

The manuscript "Trends in N2O and SF6 mole fraction in archived air samples from Cape Meares, Oregon (USA) 1978–1996" describes measurements of nitrous oxide and sulfur hexafluoride obtained from archive air samples collected in the northern hemisphere. Since there are few measurements of atmospheric N2O and SF6 in the northern hemisphere from this period, these data will provide a valuable addition to southern hemispheric archive air measurements. The analytical method is sound and the paper is well written.

Minor Comments

Page 1, Line 17:  Please consider using mixing ratio or mole fraction instead of concentration, or refer to "mole fraction in dry air" on first use of concentration.  Concentration is the amount of substance in a defined space or volume.

Page 3, Line 20:  Consider including recent papers that suggest a shorter lifetime for SF6.

Kovács, T. *et al.* (2017) 'Determination of the atmospheric lifetime and global warming potential of sulfur hexafluoride using a three-dimensional model', *Atmospheric Chemistry and Physics*, 17(2), pp. 883–898. doi: 10.5194/acp-17-883-2017.

Ray, E. A. *et al.* (2017) 'Quantification of the $SF_6$ lifetime based on mesospheric loss measured in the stratospheric polar vortex', *Journal of Geophysical Research*. doi: 10.1002/2016JD026198.

Page 4, Line 11:  Is the air dried or collected wet?

Pg .3, Line 21:  Seems like a more recent $SF_6$ mole fraction could be inserted here.  Global mean mixing ratios are available from several sources, such as the AGAGE data repository (https://agage.mit.edu/data/agage-data) or State of the Climate reports:  State of the Climate in 2017, supplement to the August 2017 issue of the Bulletin of the American Meteorological Society (BAMS Vol. 99, No. 8); (https://www.ncdc.noaa.gov/bams/2017).

Page 5. Line 7:  According to https://www.esrl.noaa.gov/gmd/ccl/refgas.html, the N2O scale associated with CB11406 (328.71 ppb) is NOAA-2006A (a 2011 update from NOAA-2006)

Page 5, Line 13:  Not sure what is meant by "sets of 6 gas analysis".  Maybe just say  "repeated analysis of a reference standard"?

Page 6, line 11:  Does the error stated here include the uncertainty on the $SF_6$ mole fraction in the dilution gas and $SF_6$ that might be present in the 1 ppm $N_2O$ aliquot?  0.001 ppt seems too small, unless you have some other way to verify $SF_6$ in the dilution gas to better than 0.001 ppt.

Page 6, Line 15: Shouldn't the slope, 0.870, be the inverse of the coefficient a1 (1.146)?  These don't quite match.

Page 10, Line 7: check spelling of "Leuker" vs "Lueker" et al. (2003)

Page 10, Line 23:  Is it known that SF6 sources are a-seasonal?  Please provide a reference.

---

## Referee Comment (RC3) · Andreas Engel (Referee) · 18 Mar 2019

Rolfe and Rice present a consistent data set of N2O and SF6 based on reanalysed whole air samples from an air archive from samples collected at Cape Meares in Oregon. This is an interesting and important study. Given that this is to a large degree a technical paper, I find that some of the details of the data retrieval could be presented more carefully. E.g. Fig 1. is not very nice and should be improved. I also have some comments with respect to the NL correction: A linear function to correct for non-linearity can only be valid over a restricted concentration range (as described by the authors). In the case of SF6 the linear correction term is applied over the entire range (from 0.59

[Figure]

-8.76 ppt) it is doubtful is this is a non-linearity effect, but rather it could be a contamination from the zero gas. In this respect, I suggest to present the NL plots (Fig 4) as difference plots, i.e. (measured – expected) vs. expected. Otherwise small deviations will not be visible and it is difficult for the reader to assess the non-linearity.

One additional importance of the paper may actually be to make this air archive better known to the scientific community, as it may be an important addition to the well-known air archive from Cape Grim in Tasmania. In so far I would also encourage the authors to provide more details on the archive itself, including possibly a note on the availability of the archive for other studies and the amount of air stored in the tanks.

Also, for such a study, a statement on data availability should be included.

I have a range of more specific comments which are partly related to the presentation and partly to the data analysis.

p.1. l. 17.: is this precision for SF6 not dependant on the mixing ratio, which has changed significantly during this time period?

p. 1. L. 29: please specify that the stability is only valid for these compounds. Other gases may be much more critical.

p.2.l. 13.: also the temporal resolution of firn samples is limited.

p. 2.l. 17.: I think it would be worthwile to add some comments here, especially mention the best known air archive, i.e. the one from Cape Grim, incl. some references to reanalysis from air archives, e.g. from Laube, Oram and Vollmer.

p.2.,l. 20: a reference to the updated trend from the most recent WMO report (chapter1) could be made here).

p.3. l. 20: please include some discussion on the recent re-evalution of the SF6 atmospheric lifetime e.g. by Ray et al.,

p.5. l., 22: I think some discussion on the reproducibility for low SF6 mixing ratios

is necessary here. Does it differ from those of the NOAA standards? What is the implication for the reanalysis?

p.6. l1ff.: Have the authors considered a cross interference between N2O and CO2? Are they separated chromatographically? If not then there could be a co-elution problem and then the dilution of the standard may result in a different matrix than in the case of air (which has shown ifferent relative trends of CO2 and N2O). Co-elution of CO2 and N2O may effect the sensitivity of the detector, which impacts the non-linearity correction.

p.6. l. 23: I think it is wrong to refer to measured N2O here; this is the "linear-response" evaluation.

p.7.l.3.: as above

p.7. l. 6: a range of values for which such a linear correction has been applied should be given here also.

Please use the newest AGAGE overall reference: Prinn et al., Earth System Sci. Data, 2018

p.8. l. 25: give years for the increase rates.

p.9. l. 13.: the ref. to Levin should be placed behind 1995.

p.9.l.17: as Edgar is largely derived from an inversion of observations, it is somewhat a circular argument to state that there is agreement.

p.9. l. 31.: this sentence sounds funny.

p.10. l. 11.: is the seasonality independent of the mixing ratios? Otherwise please give the years for which this is valid.

p.10.l24.: specify what you mean by vertical diffusion here.

p.11.l. 3: check grammar on this sentence.

---

## Referee Comment (RC4) · Anonymous Referee #3 · 21 Mar 2019

The paper on trends of N2O and SF6 from archived air samples from Cape Meares, Oregon is concise and reasonably well written. Though there are other long-term measurements of N2O in the 1980s and 1990s analysis techniques have steadily improved. The SF6 data presented is one of only a few precise time-series measurements of this important trace gas in the 1980s and early 90s. The data presented in this paper benefits from using the best modern method for measuring SF6 and N2O especially when considering the limited air in archive samples. The data and subject are certainly appropriate for publication. Please review the comments below.

General and Specific Comments

[Figure]

In some cases, the succinctness of the paper leads to some ambiguity or lack of understanding for the reader. Maybe the authors could expand on a couple of the following points or ideas.

Page 3, line 10. "Models have shown that future climate conditions will likely amplify N2O production". Expand on this thought. Why is this so?

Page 3, line 15. Add Hall et al. 2011

Page 3, line 20. Consider adding the following citation for a recent estimate of SF6 lifetime.

E. A. Ray et al., Quantification of the SF6 lifetime based on mesospheric loss measured in the stratospheric polar vortex. J Geophys Res 122, 4636-4648 (2017).

Page 5, line 4. How did you arrive at the detector temperature of 310 C? Was it optimized for N2O and or SF6?

Page 5, line 9. Maybe not necessary to the paper. Why sample the archive air 6 times and then the reference gas 6 times instead of alternating between the two types of samples? Wouldn't alternating better track signal drift from injection to injection?

Page 7, Results. Can you comment on why there were some large outliers? Problems with the sample or the integrity of a few flasks? Were the outliers the same for both N2O and SF6? Why were two different criteria for residual outliers (2-sigma for N2O and 3-sigma for SF6) used?

Page 7, line 20. It is uncertain to the reviewer what "bootstrapping residual variability 1000 times" means. Did you sample subsample the data 1000 times and re-smooth?

Page 8, line 5. You could cite Geller et al. as well.

Geller, L. S., J. W. Elkins, J. M. Lobert, A.D. Clarke, D. F. Hurst, J. H. Butler, and R. C. Myers, Tropospheric SF6: observed latitudinal distribution and trends, derived emissions and interhemispheric exchange time, Geophys. Res. Len., 24, 675-8, 1997.

Technical Corrections

Page 1, line 14. "prior to" to "before"

Page 2, line 22. "major" to "primary"

Page 4, line 20. "Peak separation is achieved by two Poropak Q 80/100 mesh columns" to "Two Poropak Q 80/100 mesh columns achieve peak separation"

Page 4, line 22. "to significantly improve baseline signal stability" to "to improve baseline signal stability significantly"

Page 5, line 20. "a two-week period" to "two weeks"

Page 6, line 4. "Error" to "The error"

Page 6, line 6. "To characterize of the" to "To characterize the" (remove the "of")

Page 6, line 8. "a N2O" to "an N2O"

Page 7, line 17. "analysis" to "the analysis"

Page 7, line 29. "Prinn et al." is missing the period

Page 9, line 12. Add a comma after Canada.

Page 10, line 19. "have amplitude" to "have an amplitude"

Page 10, line 22. "seasonal amplitude" to "the seasonal amplitude"

Page 11, line 10. "and minimum amplitude" to "and a minimum amplitude"

---

## Author Comment (AC2) · 23 Apr 2019

Response to Brad Hall

We thank reviewer #2 for the consideration of our manuscript and study for ACP and for their detailed review. We particularly appreciate the technical questions, comments, and corrections to the work and in the text. After careful consideration, we have addressed each of these points in the revised manuscript and know that this review has contributed to a stronger study and improved manuscript overall.

Here we detail changes made in our revised manuscript in order to address particular points for reviewer #2.

**Page 1, Line 17: Please consider using mixing ratio or mole fraction instead of concentration, or refer to "mole fraction in dry air" on first use of concentration. Concentration is the amount of substance in a defined space or volume.**

We have updated the manuscript to use the more correct technical language and replaced concentration with mole fraction or mixing ratio throughout the document where appropriate.

**Page 3, Line 20: Consider including recent papers that suggest a shorter lifetime for $SF_6$.**

We have revised the manuscript to use updated estimates of the $SF_6$ lifetime from Kovacs et al. (2017) and Ray et al. (2017) addressing this comment and that of reviewer #1.

**Page 4, Line 11: Is the air dried or collected wet?**

We have added clarification here that air was dried upon collection, removing a significant amount of water vapor, using a condenser-type system.

**Pg .3, Line 21: Seems like a more recent $SF_6$ mole fraction could be inserted here. Global mean mixing ratios are available from several sources, such as the AGAGE data repository (https://agage.mit.edu/data/agage-data) or State of the Climate reports: State of the Climate in 2017, supplement to the August 2017 issue of the Bulletin of the American Meteorological**

We appreciate this point that the manuscript should use more updated $SF_6$ global mean mole fraction and used a recent calculation of 9.3ppt for the NH in January, 2017 from Prinn et al. (2018).

**Page 5. Line 7: According to https://www.esrl.noaa.gov/gmd/ccl/refgas.html, the N2O scale associated with CB11406 (328.71 ppb) is NOAA-2006A (a 2011 update from NOAA-2006)**

We have updated the manuscript to reflect this change.

**Page 5, Line 13: Not sure what is meant by "sets of 6 gas analysis". Maybe just say "repeated analysis of a reference standard"?**

We have updated the manuscript here for clarity changing the language to "determined by repeated analysis of the reference standard".

**Page 6, line 11: Does the error stated here include the uncertainty on the $SF_6$ mole fraction in the dilution gas and $SF_6$ that might be present in the 1 ppm N2O aliquot? 0.001 ppt seems**

**too small, unless you have some other way to verify SF$_6$ in the dilution gas to better than 0.001 ppt.**

This is an important point that the 0.001ppt error is from the uncertainty in manometric measurement alone. Both the dilution gas and the N$_2$O aliquot may have trace SF$_6$ at levels below the detection limits of our instrumentation and would also contribute to uncertainty in the resulting SF$_6$ prepared samples. We have updated this discussion to point out that uncertainty is larger when including this consideration.

> *The maximum error (1σ) in SF$_6$ introduced from the manometric process is small (0.001 ppt) compared to measurement uncertainty. However, SF$_6$ present in either ultra-pure air dilution gas or the N$_2$O aliquot at trace levels below the detection limit of our measurement (<0.1 ppt) contribute to the uncertainty in prepared samples.*

**Page 6, Line 15: Shouldn't the slope, 0.870, be the inverse of the coefficient a1 (1.146)? These don't quite match.**

We thank the reviewer for catching this technical error here. Indeed the slope should be the inverse of the coefficient a1. The equations have been updated to give the correct values.

**Page 10, Line 7: check spelling of "Leuker" vs "Lueker" et al. (2003).**

We have updated the manuscript to include this change.

**Page 10, Line 23: Is it known that SF$_6$ sources are a-seasonal? Please provide a reference.**

We have included a reference to a global CTM modeling study by Patra et al. (2009) which simulated SF$_6$ mixing ratios and their seasonality at remote sites using emissions inventory (EDGAR) that lack seasonality. Thus the simulation, which matches seasonality well for remote sites (only), appears driven primarily by atmospheric transport. Additionally, we have been unable to find a (bottom-up or top-down) SF$_6$ emissions inventory with a significant seasonality.

---

## Author Comment (AC3) · 23 Apr 2019

Response to Andreas Engel

We thank the reviewer for the careful review of the manuscript and study and for their detailed comments. Particularly relevant in our revised document is improvements and additions to figures in the manuscript, which address a number of comments by reviewer #3. We have also added clarifying language in places and added additional detail in the manuscript where requested. Overall, after addressing critiques and questions posed by the reviewer in the revised manuscript, we feel the manuscript has improved and the study is strengthened.

Here we detail changes made in our revised manuscript in order to address particular points for reviewer #3.

**Figure 1 isn't very nice and needs to be improved.**

We do agree that the resolution in the schematic of our GC-ECD system was poor and have updated the figure output file to improve its resolution.

[Figure]

**Replace Non-Linear plots with Difference plots.**

We agree that difference plots can provide additional insight where non-linear effects are large. However, after consideration, we do feel that the measured v. expected plots shown in Fig. 4 are straightforward to interpret. For this reason, we have kept Fig. 4 as is and added additional difference plots for $N_2O$ and $SF_6$ in the supplemental documentation over the entire range measured. We also refer to these plots within the text when discussing non-linear effects.

[Figure]

**Figure S1.** Mole fraction difference from expected plots for $N_2O$ (a) and $SF_6$ (b) detector response calibration measurements. Solid black lines are 3rd-degree polynomials fit to the whole data range. For $N_2O$, 1st-degree polynomial fit (red-dashed line) is only fit to data with mole fractions expected to be greater than 295 ppb. For $SF_6$, 1st-degree polynomial fit spans the entire data range.

**Statement on data and archive availability for other studies and the amount of air stored in the tanks.**

A statement on data availability is included at the end of the manuscript that we make all $N_2O$ and $SF_6$ data available to the scientific community upon publication. The text in section 1 (page 4, line 13) states the current air pressure in archive canisters which ranges from 60-2000 kPa bar.

**p. 1. l. 17.: is this precision for $SF_6$ not dependent on the mixing ratio, which has changed significantly during this time period?**

The precision of measurement provided in the abstract and later in the section 2 (methods) is for current ambient mixing ratios and based on repeated analyses of the NOAA reference cylinder (328.71 ± 0.5 ppb $N_2O$, 8.76 ± 0.06 ppt $SF_6$). We have modified the language in the abstract to clarify this point as suggested. For archive samples, the absolute measurement precision was determined to be relatively consistent across the concentration range measured for $N_2O$ and $SF_6$ based on replicate analysis (see error bars Fig. 5). However, relative precision of measurement for samples is dependent on the mixing ratio measured for $SF_6$ because of the wide range of mixing ratios in the archive. As noted, the result of this is the largest relative uncertainty is associated with the oldest samples where $SF_6 \leq 1$ ppt. Text in the main body now includes:

> *Mean measurement uncertainty (1σ) of OHSU-PSU air archive samples for $N_2O$ is 0.23%. Mean measurement uncertainty (1σ) of $SF_6$ in the OHSU-PSU air archive samples ranges between 6.5% for samples below 1 ppt and 2.5% for samples at 4 ppt.*

**P. 1. L. 29.: please specify that the stability is only valid for these compounds. Other gases may be much more critical.**

We have updated the manuscript to include this change.

**P. 2. L. 13.: also the temporal resolution of firn samples is limited.**

We have updated the manuscript to include this change.

**P. 2. L. 17.: I think it would be worthwile to add some comments here, especially mention the best known air archive, i.e. the one from Cape Grim, incl. some references to reanalysis from air archives, e.g. from Laube, Oram and Vollmer.**

We agree with the reviewer and have updated the manuscript to include this change:

*The most well-known air archive is that of Cape Grim, Tasmania (41° S, 145° E) in the southern hemisphere, containing samples dating back to 1978 (Vollmer et al. 2018).*

**P. 2. L. 20.: a reference to the updated trend from the most recent WMO report (chapter1) could be made here).**

We agree with the reviewer and have updated the manuscript to include this change:

*The global mean mixing ratio of $N_2O$ in 2017 was 329.8 ppb with a mean annual trend of 0.85 ppb $yr^{-1}$ over the last 20 years (Dlugokencky et al. 2018).*

**P. 3. L. 20.: please include some discussion on the recent re-evalation of the $SF_6$ atmospheric lifetime e.g. by Ray et al. 2017.**

We have updated the manuscript to include this change.

**P. 5. L. 22.: I think some discussion on the reproducibility for low $SF_6$ mixing ratios is necessary here. Does it differ from those of the NOAA standards? What is the implication for the reanalysis?**

In terms of a study of the reproducibility at lower $SF_6$ mixing ratios, we were unable to perform a longer-term multi-day analyses as we did not have a sample of sufficient volume to do so (at low $SF_6$ mixing ratio). However, as the reproducibility tests at ambient $SF_6$ mixing ratios are very compatible with our analysis of the precision of measurement, we have no real reason to suspect the same study at lower mixing ratios would yield different results.

**P. 6. l1ff.: Have the authors considered a cross interference between $N_2O$ and $CO_2$? Are they separated chromatographically? If not then there could be a co-elution problem and then the dilution of the standard may result in a different matrix than in the case of air (which has shown different relative trends of $CO_2$ and $N_2O$). Co-elution of $CO_2$ and $N_2O$ may effect the sensitivity of the detector, which impacts the non-linearity correction.**

Our tests (using NDIR) indicate that $CO_2$ is well separated from $N_2O$ on a 5.5m Porapak Q column.

**P. 6. L. 23.: I think it is wrong to refer to measured $N_2O$ here; this is the "linear-response" evaluation.**

We have updated the manuscript to state "response evaluated" instead of "measured".

**P. 7. L. 3.: as above**

We have updated the manuscript to state "response evaluated" instead of "measured".

**P. 7. L. 6.: a range of values for which such a linear correction has been applied should be given here also. Please use the newest AGAGE overall reference: Prinn et al., Earth System Sci. Data, 2018**

We have updated the manuscript to include the range in values for which the linear corrections have been applied (with corrected values ranging between 298.9 – 314.8 ppb for $N_2O$ and 0.6 – 4.3 ppt for $SF_6$).

**P. 8. L. 25.: give years for the increase rates.**

Years for the rate of increase the growth rates are calculated over are provided in the first sentence of the paragraph (1978 – 1996). However, we have added "over this same time period" to the end of the sentence to improve clarity.

**P. 9. L. 13.: the ref. to Levin should be placed behind 1995.**

We have updated the manuscript to include this change.

**P. 9. L. 17.: as Edgar is largely derived from an inversion of observations, it is somewhat a circular argument to state that there is agreement.**

In the case of $SF_6$, the reviewer makes a good point here that Edgar emissions inventories are based in-part of inversion of atmospheric measurements. However, atmospheric $SF_6$ data are particularly sparse in 1980s and early 1990s, particularly in the northern hemisphere where an overwhelming majority of emissions occur. Thus, the agreement between middle latitude northern hemisphere Cape Meares, OR (45N) data and EDGAR emissions inventory is useful for updating and improving emissions inventories.

**P. 9. L. 31.: this sentence sounds funny.**

We have restructured the sentence to the following:

*Other mid-latitude northern hemisphere sites also show a seasonal phase similar to that observed at Cape Meares.*

**P.10. L. 11.: is the seasonality independent of the mixing ratios? Otherwise please give the years for which this is valid.**

So far as can be determined statistically, the seasonality observed at Cape Meares is independent of the mixing ratio.

**P. 10. L. 24.: specify what you mean by vertical diffusion here.**

We have updated the manuscript to not include vertical diffusion in the list of potential factors that may influence the observed seasonality at a location due to the ambiguity of the term.

**p.11.l. 3: check grammar on this sentence.**

We have updated the manuscript to the following:

*The analysis of archived air samples gives the mole fraction of $N_2O$ in 1980 to be $301.5 \pm 0.3$ ppb ($1\sigma$), rising to $313.5 \pm 0.3$ ppb ($1\sigma$) in 1996.*

---

## Author Comment (AC4) · 23 Apr 2019

Response to Anonymous Referee #3

We thank the reviewer for the careful review of the manuscript and study and for their detailed comments. Their critiques, especially their attention to technical details, have helped to significantly strengthen our manuscript.

Here we detail changes made in our revised manuscript and responses to questions addressed by Anonymous Referee #3.

**General and Specific Comments**

**Page 3, line 10. "Models have shown that future climate conditions will likely amplify $N_2O$ production". Expand on this thought. Why is this so?**

It is estimated that 44-73% of $N_2O$ emissions originate from land ecosystems (Hirsch et al. 2006; Davidson et al. 2009). A warmer climate will most likely enhance these emissions (Arneth et al. 2010), driving a positive feedback loop. Stocker et al. (2013) investigates the role these feedback loops play in future climate conditions. We have updated the text to the following:

*Models have shown that future climate conditions will likely amplify $N_2O$ production through positive climate feedback effects, meaning a linear increase in time may under-predict future concentrations based on the current rate of change (Khalil and Rasmussen 1983; Stocker et al. 2013).*

**Page 3, line 15. Add Hall et al. 2011**

We have updated the manuscript to reflect this change.

**Page 3, line 20. Consider adding the following citation for a recent estimate of $SF_6$ lifetime.**

We have revised the manuscript to use updated estimates of the $SF_6$ lifetime from Kovacs et al. (2017) and Ray et al. (2017).

**Page 5, line 4. How did you arrive at the detector temperature of 310 C? Was it optimized for $N_2O$ and or $SF_6$?**

The detector temperature of 310°C was used because we found that the response of the $N_2O$ and $SF_6$ peaks were well defined at this temperature. Chromatogram output was optimized around $N_2O$ while maintaining a robust $SF_6$ peak. No significant difference in detector response was found to using a detector temperature of 340°C reported in Hall et al. (2007) and Hall et al. (2011).

**Page 5, line 9. Maybe not necessary to the paper. Why sample the archive air 6 times and then the reference gas 6 times instead of alternating between the two types of samples? Wouldn't alternating better track signal drift from injection to injection?**

The procedure of running 6 measurements in a row of a sample or standard instead of alternating between standard and sample measurements was used because it was determined that while measuring $N_2O$, if a run contained an outlier (greater than $2\sigma$), it was statistically more likely to be the first measurements (~30% of the time) compared to being in another position (~15% of the time). Outlier probability was evenly spread across measurement position for $SF_6$ (~17% for outlier

to be in any position). The cause of this discrepancy for measurements of $N_2O$ is unlikely to be contamination from a previous run as we are purging the sample loop (10 ml) with 9-times the sample loop volume (60 ml min$^{-1}$ for 1.5 minutes). This memory effect is mitigated when running 6 measurements back to back. Drift in the detector response over a set of 6 measurements (~50 min) is assumed to be linear.

**Page 7, Results. Can you comment on why there were some large outliers? Problems with the sample or the integrity of a few flasks? Were the outliers the same for both $N_2O$ and $SF_6$? Why were two different criteria for residual outliers (2-sigma for $N_2O$ and 3-sigma for $SF_6$) used?**

Outliers in the OHSU-PSU air archive are possibly due to several factors including storage integrity or possible contamination during collection. It is unclear what exactly caused each of the far outliers evaluated.

For the initial filtering process, far outliers in the OHSU-PSU air archive were considered to be 6*MAD (median absolute deviation) for $N_2O$ (removes 6 samples) and 7*MAD for $SF_6$ (removes 2 samples). The far outliers for $N_2O$ are not the far outliers for $SF_6$. 7*MAD is used for $SF_6$ as opposed to 6*MAD because the annual increase in measured values for $SF_6$ (~10% yr$^{-1}$) is significantly larger than in $N_2O$ (~0.25% yr$^{-1}$). It is important to note that roughly half of the 159 samples measured from the OHSU-PSU air archive date prior to 1985, meaning the median measured $SF_6$ mole fraction will be biased towards this early period. By using 7*MAD for $SF_6$, we only remove values that clearly lie outside of a reasonable measurement.

For the second filter, Polynomial fits (1$^{st}$ degree for $N_2O$ and 2$^{nd}$ degree for $SF_6$) were applied to the data and residuals outside of 2σ for $N_2O$ and 3σ for $SF_6$ were removed. 3σ was used for $SF_6$ because the data points fit tightly to the polynomial fit. The second filter removed another 6 samples for $N_2O$ (for a total of 12) and another 2 samples for $SF_6$ (for a total of 4). Again, the outliers removed for $N_2O$ are not the same samples removed for $SF_6$. We found that using 2σ for $SF_6$ removed data points unnecessarily from the analysis.

**Page 7, line 20. It is uncertain to the reviewer what "bootstrapping residual variability 1000 times" means. Did you sample subsample the data 1000 times and re-smooth?**

The bootstrap process consists of calculating a new value for each data point from a normal distribution with mean equal to the measurement and standard deviation equal to the residual variability found from the original LOWESS regression. We repeat the LOWESS regression calculation for the new data set. The process is completed 1000 times, from which we calculate the 95% confidence interval for the original LOWESS regression to the measured data.

**Page 8, line 5. You could cite Geller et al. as well**

We have updated the manuscript to reflect this change.

**Technical Corrections**

**Page 1, line 14. "prior to" to "before"**

We have updated the manuscript to reflect this change.

**Page 2, line 22. "major" to "primary"**

We have updated the manuscript to reflect this change.

**Page 4, line 20. "Peak separation is achieved by two Poropak Q 80/100 mesh columns" to "Two Poropak Q 80/100 mesh columns achieve peak separation"**

We have updated the manuscript to reflect this change.

**Page 4, line 22. "to significantly improve baseline signal stability" to "to improve baseline signal stability significantly"**

We have updated the manuscript to reflect this change.

**Page 5, line 20. "a two-week period" to "two weeks"**

We have updated the manuscript to reflect this change.

**Page 6, line 4. "Error" to "The error"**

We have updated the manuscript to reflect this change.

**Page 6, line 6. "To characterize of the" to "To characterize the" (remove the "of")**

We have updated the manuscript to reflect this change.

**Page 6, line 8. "a N2O" to "an N2O"**

The sentence has been reworded to:

> *To properly account for this interference, $SF_6$ dilutions at low mixing ratios (0.6 - 6.0 ppt) must have $N_2O$ mole fractions that reflect expected mole fractions in archived samples (300 - 315 ppb).*

**Page 7, line 17. "analysis" to "the analysis"**

We have updated the manuscript to reflect this change.

**Page 7, line 29. "Prinn et al." is missing the period**

We have updated the manuscript to reflect this change.

**Page 9, line 12. Add a comma after Canada.**

We have updated the manuscript to reflect this change.

**Page 10, line 19. "have amplitude" to "have an amplitude"**

We have updated the manuscript to reflect this change.

**Page 10, line 22. "seasonal amplitude" to "the seasonal amplitude"**

We have updated the manuscript to reflect this change.

**Page 11, line 10. "and minimum amplitude" to "and a minimum amplitude"**

We have updated the manuscript to reflect this change.

---

## Author Response (AR1)

**Rolfe and Rice 2019 Referee Edits**

5

20

We would like to thank all of the reviewers for their thorough read of our manuscript and careful consideration of our study. In particular, we appreciate the point the reviewers make on comparison of our new dataset with  $N_2O$  and  $SF_6$  datasets in the prior literature, technical questions, comments, and corrections to the work and in the text. We appreciate the technical corrections and believe this review process has helped to strengthen our manuscript.

Here we detail changes made in our revised manuscript in order to address particular points for Eric Ray.

10 Pg. 1, line 25: I think you mean 0.04 ppt rather than ppb.

We have revised the manuscript to correct this typographic error.

Pg. 3, line 19-20: There are two more recent studies on the lifetime of SF6 that should be included here since they both significantly reduce the estimated lifetime, Ray et al., JGR, 2016 and Kovacs et al., ACP, 2017.

15 We have revised the manuscript to use updated estimates of the SF6 lifetime from Kovacs et al (2017) and Ray et al. (2017) addressing this comment and that of reviewer #1.

**Pg. 6, line 12: Change 'provides' to 'provide'.**

We have revised the manuscript to correct this typographic error.

Pg. 7, line 21: Even though it's apparent from the values of the concentration you should add 'N2O of' before '301.5' since the figure includes both N2O and SF6.

We have revised the manuscript to change language to "N2O mixing ratio of...".

Section 3.1: You mention comparable measurements and their locations in the text of this section but it would be easier to see this information in a figure. What would be useful is a plot of concentration vs. latitude at two different times, one at the beginning of your measurement time

25 series and one at the end. By including all available surface measurements, it will be easy to see how many other measurements exist for each time and how it changed. Since the concentrations changed enough over the period of your measurements you could just color the two different times differently and they will fit on the same plot. The lack of measurements in the 1970s should be readily apparent from a plot of this type.

While we find adding raw data from prior studies makes Fig. 5 unnecessarily busy and detracts from this new contribution of Cape Meares data to the atmospheric community, we can see that a visual

5 comparison of our results with prior work is useful in addition to the discussion we already have in the manuscript. Additionally, we note that much of the raw data for SF6 previously published is not available through open access WDCGG (or other platforms). To address this, we have included a supplemental figure which compares regressed fits through the Cape Meares dataset with fits from other comparator sites in the literature. We hope this will help reader assess how this new dataset fits within historical published trends in N-O and SE.

10 historical published trends in  $N_2O$  and  $SF_6$ .

Figure S2. 3-year LOWESS regressions of measurements of mole fraction versus date of collection, N2O (a) and SF6 (b). Station codes: CMO = Cape Meares, Oregon, USA, NWR = Niwot Ridge, Colorado, USA, MHD = Mace Head, Ireland, THD = Trinidad Head, California, USA, CGO = Cape Grim, Tasmania, ALT = Alert, Canada. N2O data sources:
15 Atmospheric Lifetime Experiment (ALE, now AGAGE), Massachusetts Institute of Technology, Building 54-1312 Cambridge, MA 02139-2307, https://agage.mit.edu/; Global Atmospheric Gases Experiment (GAGE, now AGAGE), Massachusetts Institute of Technology, Building 54-1312 Cambridge, MA 02139-2307, https://agage.mit.edu/; National Oceanic and Atmospheric Association / Earth System Research Laboratory (NOAA/ESRL), 325 Broadway Boulder, CO 80305-3337, http://www.cmdl.noaa.gov/index.html; Advanced Global Atmospheric Gases Experiment Science Team
20 (AGAGE), Massachusetts Institute of Technology, Building 54-1312 Cambridge, MA 02139-2307, https://agage.mit.edu/;

N2O data collected from World Data Center for Greenhouse Gases (WDCGG) https://gaw.kishou.go.jp/. SF6 data is digitized from plots in Rigby et al. 2010 and Levin et al. 2010.

Pg. 10, lines 23-25: Also, seasonal transport from the stratosphere can influence SF6 due to the high growth rates, especially in these early years. Growth rates of  $\sim 10\%/yr$  means that

5 stratospheric air with a mean age of 2 years will have ~20% lower concentrations compared to tropospheric values. For example, the seasonal cycle of CFCs have a minimum in the summer of each hemisphere due to the transport of relatively low concentrations due to photochemical destruction (e.g. Liang et al., JGR, 2008).

We thank the reviewer for this valid point and have updated the final paragraph of section 3.3 to address

10 it specifically:

Seasonal transport from STE adds relatively depleted  $SF_6$  air into the troposphere from the stratosphere. The seasonal phase of  $SF_6$  observed at Cape Meares closely reflects seasonality phasing observed in CFCs in the northern hemisphere driven by STE (Liang et al. 2008). Modeling atmospheric transport effects on  $SF_6$  at Cape Meares could help confirm amplitude and phase

15 *reported here.*

Here we detail changes made in our revised manuscript in order to address particular points for Brad Hall.

Page 1, Line 17: Please consider using mixing ratio or mole fraction instead of concentration, or

20 refer to "mole fraction in dry air" on first use of concentration. Concentration is the amount of substance in a defined space or volume.

We have updated the manuscript to use the more correct technical language and replaced concentration with mole fraction or mixing ratio throughout the document where appropriate.

**Page 3, Line 20: Consider including recent papers that suggest a shorter lifetime for SF6.**

25 We have revised the manuscript to use updated estimates of the SF6 lifetime from Kovacs et al. (2017) and Ray et al. (2017) addressing this comment and that of reviewer #1.

Page 4, Line 11: Is the air dried or collected wet?

We have added clarification here that air was dried upon collection, removing a significant amount of water vapor, using a condenser-type system.

Pg .3, Line 21: Seems like a more recent SF6 mole fraction could be inserted here. Global mean mixing ratios are available from several sources, such as the AGAGE data repository

5 (https://agage.mit.edu/data/agage-data) or State of the Climate reports: State of the Climate in 2017, supplement to the August 2017 issue of the Bulletin of the American Meteorological
 We appreciate this point that the manuscript should use more updated SF6 global mean mole fraction and used a recent calculation of 9.3ppt for the NH in January, 2017 from Prinn et al. (2018).

Page 5. Line 7: According to https://www.esrl.noaa.gov/gmd/ccl/refgas.html, the N2O scale associated with CB11406 (328.71 ppb) is NOAA-2006A (a 2011 update from NOAA-2006)

We have updated the manuscript to reflect this change.

Page 5, Line 13: Not sure what is meant by "sets of 6 gas analysis". Maybe just say "repeated analysis of a reference standard"?

We have updated the manuscript here for clarity changing the language to "determined by repeated analysis of the reference standard".

Page 6, line 11: Does the error stated here include the uncertainty on the SF6 mole fraction in the dilution gas and SF6 that might be present in the 1 ppm N2O aliquot? 0.001 ppt seems too small, unless you have some other way to verify SF6 in the dilution gas to better than 0.001 ppt.

This is an important point that the 0.001ppt error is from the uncertainty in manometric measurement 20 alone. Both the dilution gas and the N2O aliquot may have trace SF6 at levels below the detection limits of our instrumentation and would also contribute to uncertainty in the resulting SF6 prepared samples. We have updated this discussion to point out that uncertainty is larger when including this consideration.

25

15

The maximum error  $(1\sigma)$  in SF6 introduced from the manometric process is small (0.001 ppt) compared to measurement uncertainty. However, SF6 present in either ultra-pure air dilution gas or the N2O aliquot at trace levels below the detection limit of our measurement (<0.1 ppt) contribute to the uncertainty in prepared samples.

**Page 6, Line 15: Shouldn't the slope, 0.870, be the inverse of the coefficient a1 (1.146)? These don't quite match.**

We thank the reviewer for catching this technical error here. Indeed the slope should be the inverse of the coefficient a1. The equations have been updated to give the correct values.

5 Page 10, Line 7: check spelling of "Leuker" vs "Lueker" et al. (2003). We have updated the manuscript to include this change.

**Page 10, Line 23: Is it known that SF6 sources are a-seasonal? Please provide a reference.**

We have included a reference to a global CTM modeling study by Patra et al. (2009) which simulated SF6 mixing ratios and their seasonality at remote sites using emissions inventory (EDGAR) that lack

10 seasonality. Thus the simulation, which matches seasonality well for remote sites (only), appears driven primarily by atmospheric transport. Additionally, we have been unable to find a (bottom-up or topdown) SF6 emissions inventory with a significant seasonality.

**Here we detail changes made in our revised manuscript in order to address particular points for**

15 Andreas Engel.

**Figure 1 isn't very nice and needs to be improved.**

We do agree that the resolution in the schematic of our GC-ECD system was poor and have updated the figure output file to improve its resolution.

**Replace Non-Linear plots with Difference plots.**

We agree that difference plots can provide additional insight where non-linear effects are large. However, after consideration, we do feel that the measured v. expected plots shown in Fig. 4 are straightforward to interpret. For this reason, we have kept Fig. 4 as is and added additional difference plots for  $N_2O$  and  $SF_6$  in the supplemental documentation over the entire range measured. We also refer to these plots within the text when discussing non-linear effects.